# A Circle-Drawing Task for Studying Reward-Based Motor Learning in Children and Adults

**DOI:** 10.3390/bs14111055

**Published:** 2024-11-06

**Authors:** Nina M. van Mastrigt, Jeroen B. J. Smeets, A. Moira van Leeuwen, Bernadette C. M. van Wijk, Katinka van der Kooij

**Affiliations:** 1Department of Human Movement Sciences, Vrije Universiteit Amsterdam, 1081 HV Amsterdam, The Netherlands; j.b.j.smeets@vu.nl (J.B.J.S.); a.m.van.leeuwen@vu.nl (A.M.v.L.); b.c.m.van.wijk@vu.nl (B.C.M.v.W.); k.vander.kooij@vu.nl (K.v.d.K.); 2Department of Psychology, Justus-Liebig-Universität Gießen, 35394 Gießen, Germany; 3Department of Neurology, Amsterdam University Medical Centers, Amsterdam Neuroscience, University of Amsterdam, 1105 AZ Amsterdam, The Netherlands

**Keywords:** motor learning, adaptation, reward, development

## Abstract

Childhood is an obvious period for motor learning, since children’s musculoskeletal and nervous systems are still in development. Adults adapt movements based on reward feedback about success and failure, but it is less established whether school-age children also exhibit such reward-based motor learning. We designed a new ‘circle-drawing’ task suitable for assessing reward-based motor learning in both children (7–17 years old) and adults (18–65 years old). Participants drew circles with their unseen hand on a tablet. They received binary reward feedback after each attempt based on the proximity of the average radius of their drawing to a target radius set as double the radius of their baseline drawings. We rewarded about 50% of the trials based on a performance-dependent reward criterion. Both children (10.1 ± 2.5 (mean ± SD) years old) and adults (37.6 ± 10.2 years old) increased the radius of their drawings in the direction of the target radius. We observed no difference in learning between children and adults. Moreover, both groups changed the radius, less following reward than following reward absence, which is a sign of reward-based motor learning. We conclude that school-age children, like adults, exhibit reward-based motor learning.

## 1. Introduction

In the fields of rehabilitation [1,2] and education [3], there is interest in applying neuroscientific insights in the design of training exercises. Knowing how to optimize feedback in training exercises could, for example, enhance learning and motivation [4]. An important form of feedback is reward feedback, signaling the success or failure of a movement attempt. Such feedback can influence task motivation and enjoyment [5,6] and also provides information for ‘reward-based’ motor learning [7,8]. Although childhood is an obvious period for motor learning, neuroscience is only starting to uncover reward-based motor learning in ages younger than the university student population tested [9,10]. This area of research is important, because adult learning might not be a good model of how younger age groups learn. After all, brains and musculoskeletal systems are still in development [11] up to adulthood. Hence, the aim of this study is to establish a task for studying reward-based motor learning in children. For simplicity, we refer to the entire age group younger than 18 as ‘children’, laying the groundwork for future studies that can assess reward-based motor learning in subgroups such as childhood (3–11 years) and adolescence (12–18 years). Establishing a novel task is necessary, because existing tasks designed to study motor learning might be boring or not intuitive for children [12]. When children do not understand the task or stop attending to the task because they are bored, this might confound the potential effects of age on learning and reduce the measurement sensitivity.

In reward-based motor learning, adults learn by repeating movements following reward (signaling success) and varying movements following reward absence (signaling failure) (e.g., [13,14]). A characteristic of such learning is exploration, as reflected by larger changes in movements following reward absence than following a reward [15]. In young adults, reward-based motor learning has been observed in various simple target-directed reaching tasks and trajectory-matching tasks in which arm movements are registered with a robot arm or a handheld stylus on a drawing tablet, and digital reward feedback signaling success or failure is delivered on a screen [8,13,16,17,18,19]. Participants typically learn by finding or following an invisible reward zone that is abruptly or gradually shifted away from a visual target, meaning that they learn to reach to a different place than where they see this visual target. Both implicit and explicit processes contribute to such reward-based motor learning [14]. While simple tasks can be learned in this way, the effectiveness of reward-based motor learning decreases with task complexity, as the mapping between control parameters and feedback becomes many-to-one, requiring the learner to solve a credit assignment problem. For instance, learning decreases with the number of target movements and with the number of spatial dimensions the reward feedback is based on [19].

We expect children to show reward-based motor learning. Childhood is an obvious period for motor learning, and reward-based motor learning is, in addition to error-based learning, one of the key mechanisms for motor learning. Reward-based motor learning can be described with a simple mechanism [13] and has been attributed to brain structures such as the basal ganglia and motor cortex [20], that mature relatively early in childhood, i.e., preceding the maturation of higher-order brain areas such as the frontal cortex [11]. Moreover, reward-based motor learning can happen implicitly [14]. In fact, after data collection, our expectation that children can exhibit reward-based motor learning based on binary success-failure feedback was confirmed: in a preprint, Hill and colleagues reported reward-based learning of a reaching endpoint in children of three years and older [9].

While we expect children to show reward-based motor learning, their learning may be less than adults due to differences in the development of motor control and cognition. Explicit processes such as decision-making are attributed to the frontal cortex [21], which develops during adolescence [11]. These explicit processes might play an important role in reward-based motor learning [21,22], and hence, learning may increase with age. In addition, motor control is important in reward-based motor learning: highly precise movements due to low motor noise variability and high variability due to exploration both facilitate reward-based motor learning [13,23]. Children have demonstrated higher levels of motor variability, which might be attributed to motor noise [24,25,26] and might show reduced reward-based learning. Moreover, children have been shown to tune variability less to the task-relevant dimensions where exploration is beneficial [27,28], which might hamper their reward-based learning as compared to adults.

The aim of this study is to test the hypothesis that children, like adults, exhibit reward-based motor learning. To make sure that we could measure reward-based motor learning in children, we developed a reward-based circle-drawing task with reward feedback based on the size of a drawn circle. In this task, participants draw invisible circles with their unseen hand on a drawing tablet with a digital pen. A reward can be obtained by correctly adjusting the circle size based on binary reward feedback on the circle size after each attempt, signaling success or failure. This circle-drawing task is based on trajectory tasks reported earlier [17,18] that study learning without perturbing feedback. In these tasks, adults drew a line from a central starting position and were rewarded based on the curvature and direction of a drawn trajectory. A downside of such trajectory tasks is that, after drawing the trajectory, the participant has to return the hand to the starting position. When drawing circles rather than curves, the hand is naturally returned to the starting position. In addition, participants receive instructions embedded in a game narrative. By increasing usability this way, we aimed to establish an intuitive task that can measure reward-based motor learning to the same extent in children and adults. Additional benefits of the task are that reward feedback is veridical (i.e., not perturbed) and that task complexity can be easily manipulated by basing reward on a single or multiple characteristics of the drawn circle. We predicted that both children (7–17 years old) and adults (18–65 years old) would adapt the size of the circle they draw to the reward feedback. Furthermore, we predicted larger changes in the size of the circle following reward absence compared to following reward presence, reflecting exploration, a key ingredient in reward-based motor learning.

## 2. Materials and Methods

The experimental design and statistical tests are pre-registered on As Predicted.org: https://aspredicted.org/ds7tx.pdf (accessed on 21 December 2023). In the description of the methods below, we mention where we deviate from this pre-registration and motivate why we did so.

### 2.1. Participants

The experiment was performed at the NEMO Science Museum in Amsterdam, the Netherlands, as part of their Science Live program, which aims to let researchers collect data among museum visitors and to let museum visitors experience the process of science. Dutch-speaking and English-speaking museum visitors aged 7 to 65 years old were recruited for a drawing task by banners and flyers in the museum. We informed the museum visitors that the purpose of the study was to study how we learn to move and that participants would be doing a 20-min drawing task on a computer. Participants were included or excluded based on age, on language, and on the presence of a parent or a legal guard to provide consent for participation for participants aged 7 until 14. All participants provided informed consent before their voluntary participation. For participants below the age of 15, this consent was provided by a parent or legal guardian. For participants aged 15 to 17, both the child and parent had to provide consent. The protocol was approved by the Institutional Ethical Review Board of the Faculty of Behavioral and Movement Sciences of the Vrije Universiteit Amsterdam (VCWE-2023-100R3).

Of the 128 participants who participated, we included 100 participants in the data analysis: 69 children (28 male, 37 female, 4 other; 10 left-handed; reported rounded-down age [29] 10.1 ± 2.5 (mean ± SD) years) and 31 adults (18 male, 12 female, 1 other; 5 left-handed; reported rounded-down age 37.6 ± 10.2 years) (Figure 1b). We included 48 Dutch-speaking participants (28 children) and 52 English-speaking participants (41 children). The remaining 28 datasets were not included in the analysis. Some datasets were not complete due to a bug in the software for the experimental task, which caused computer crashes independent of participants’ drawing behavior (26 participants). Other datasets were not included in the analysis, because participants appeared to not have drawn circles (2 participants, see Section 2.3). After the first day of testing the software bug was resolved.

### 2.2. Procedure and Experimental Task

Participants performed a circle-drawing task with binary reward feedback on circle size (Appendix A). The drawing was performed with a digital pen on a drawing tablet (WACOM Intuos Medium, 27.5 cm × 21.7 cm, 200 Hz) (Figure 1a) as part of a custom-programmed circle-drawing task (in Unity3D) viewed on a laptop monitor. Whereas a normal pen leaves a visible trace, in this experiment, participants received no visual information on the drawn trajectories.

The experimental procedure was standardized (available at https://osf.io/j6752, accessed on 8 October 2024), meaning that all participants received the same verbal instructions. Participants took place behind a desk with the laptop and Wacom device. To make the task intuitive for children, the instructions about the task were embedded in a story. Participants were told they would be seeing a bear on screen (Figure 1a). As the bear had no nose, their task would be to draw a circular nose for him. Participants were instructed to make the bear happy by adjusting the size of the circles they drew. The experimenter stressed that the location of the circle on the tablet did not matter in making the bear happy.

A trial started once the participant put the pen on the tablet and ended once the participant lifted the pen for longer than 500 milliseconds. Participants could not see the circle they drew, but after every attempt, they received binary reward feedback (success or failure) based on the drawn circle. Participants were free to choose their drawing speed and, on average, took about 2 s to draw a circle. After five trials in which the reward was based on a target radius of 4.3 cm, we set a target radius that was double the average drawn radius on the first five trials (Figure 1c). Performance was rewarded based on the ratio between the drawn radius and this target radius. If this ratio is one, the error is zero. To handle circle drawing errors caused by too small (a ratio smaller than one) and too large radii (a ratio higher than one) in the same way, we calculated a relative radius error. We defined the relative radius error as the maximum of the ratio between the radius of the drawn circle and that of the target circle and its inverse minus one. This way, the relative radius error is positive when the drawn circles are either too small or too large and zero when the drawn radius is equal to the target radius. When participants received positive reward feedback, the bear would show a happy face, accompanied by a positive ‘bell’ sound. When participants received negative feedback, the bear would show a sad face accompanied by a negative ‘buzzer’ sound (Figure 1a). The reward feedback was based on an adaptive reward criterion using the past ten values for the relative radius error (Figure 1d). This approach allowed us to sort the past ten errors from small to large, handling errors caused by too small and too large radii in a similar way. The reward criterion was then set such that, in the next trial, errors smaller than the fifth-largest error (close to the median) would be rewarded. For the first ten trials, the available trials up to that point were used. This criterion was designed to keep participants’ success frequency over the past ten trials around 50%, allowing us to analyze how participants adapted their variability to the reward feedback [30].

After the instruction, the experimenter handed the pen to the writing hand of the participant. The participant could practice drawing a circle on the tablet five times. Afterward, the participant was asked to explain what they had been doing and how they had interpreted the feedback to check whether they had understood the instructions. Lastly, participants put on headphones to make sure the feedback was private, and their writing hand and the tablet were covered by a curtain. This was to prevent participants from obtaining visual information about the circle size.

The main phase of the experiment consists of 80 circle-drawing trials described above. To collect a behavioral motivation measure, we added a second phase (motivation phase), in which participants could continue the task until they felt like stopping the experiment. The experimenter sat opposite the participant and instructed the participant to raise their hand once they had finished the circle-drawing task as instructed on the screen, which would be after the first phase. Once the participant raised their hand, the experimenter informed them that she had collected enough data and that she needed some time to complete some forms. She told the participant that they could choose to continue playing the drawing game in the meantime by pressing a button on the laptop or to stop playing the game by pressing another button that was visible after each trial.

To obtain a self-reported measure of motivation, after finishing both phases of the task, participants filled in a modified version of the Intrinsic Motivation Inventory [31,32]. The five items represent on a five-point scale task enjoyment, perceived self-competence, perceived effort/importance, perceived tension, and the motivation to continue. Items were phrased as “I would like to play this game again.”, “I enjoyed playing this game.”, “I was good at this game.”, “I tried my best to score as many points as possible.”, and “I felt nervous while I was playing this game.”.

### 2.3. Data Analysis

Each trial consisted of one attempt to draw a circle, followed by binary feedback about the success or failure based on the radius ratio. The tablet recorded the drawn trajectories with a sampling frequency of 60 Hz. To quantify the size of the drawing, we started by determining its center as the mean of the maximum and minimum horizontal and vertical positions. As participants could vary their drawing velocity during a trial, the drawn trajectory, which was sampled at 60 Hz, was spatially resampled into 50 data points, such that we could define the radius as the average Euclidian distance between each spatially resampled data point on the trajectory and its center (Appendix B
Figure A1). This way, the radius was not biased by distances to the center in parts of the drawing that were performed at a low speed.

Since we study learning, it is essential to include all trials of a participant. If a participant’s drawings did not resemble circles at all, we excluded the participant from the data analysis. A perfect circle has an aspect ratio of one and has zero distance between its start and endpoint. For every trial, we determined the aspect ratio as the ratio between two perpendicular intersections of the drawn trajectory that deviated the most from one. If this aspect ratio was smaller than 0.1 or larger than 10 or if the trajectory length was smaller than twice the distance between the start and endpoint of the trajectory, we regarded the drawing as non-circular. A participant was excluded from the analysis if more than 20% of the trials were non-circular. For other participants, all trials were included in the analysis.

#### 2.3.1. Learning

We measured learning by the fraction learned: the ratio between the average radius of the last ten trials of the main phase and the baseline radius minus one. As the target ratio is twice the baseline ratio, this definition results in a value of zero for baseline performance (no learning) and a value of one for the target radius (perfect learning). Values below zero indicate final main phase radii that are smaller than the baseline radius, and values above one indicate final main phase radii that are larger than the target radius.

#### 2.3.2. Variability

The quantification of variability is based on trial-to-trial changes in the drawn radius. As we are interested in relative changes, for trial-to-trial changes, we use the ratio between the radius on trial t+1 and the radius on trial t-1. We use trial t-1 rather than trial t when comparing changes following rewarded trials and following non-rewarded trials to prevent sampling biases induced by the presence of performance constraints defined by the reward zone on rewarded trials that are not present on non-rewarded trials [30]. As we are interested in the size of the trial-to-trial change and not the direction (‘halving or doubling’), a trial-to-trial change reflecting doubling the radius should be interpreted in the same way as a trial-to-trial change in which the radius is halved. To this end, we defined the trial-to-trial ratio as the maximum of the trial-to-trial change and its inverse, yielding trial-to-trial ratios larger than one only. Variability was quantified as the median trial-to-trial ratio in the radius of the drawing. Although exploration could be formalized as the difference in variability following non-rewarded and rewarded movements, in the current study, we did not quantify exploration due to the low number of trials [30].

#### 2.3.3. Motivation

In addition to the primary outcome measures reported above, we explored whether differences between children and adults could be explained by different motivations for the task. We measured both self-reported motivation, quantified by the mean score of the five items of the modified Intrinsic Motivation Inventory, ranging between 1 (low motivation) and 5 (high motivation), and a behavioral measure of motivation: the number of trials completed in the motivation phase of the experiment, ranging between 0 and 20.

We pre-registered the behavioral measure of motivation as our primary measure of motivation. However, following the study, the experimenters indicated that they had trouble providing the instructions for the motivation phase of the experiment in a reliable manner for all participants. In addition, we expected the behavioral measure to show a wide distribution with values between zero and twenty trials but instead observed a bimodal distribution with participants either stopping the task directly after the main phase (i.e., zero trials) or fully completing the motivation phase (i.e., twenty trials). We therefore inspected the internal consistency of the Intrinsic Motivation Inventory by computing Cronbach’s alpha and item–item correlations for participants with complete question sets. We found a low internal consistency of α = 0.56 (N_complete_ = 71), with item–item correlations ranging between 0 and 0.55. We therefore decided to analyze motivation as the mean of the two questions that correlated most strongly (r = 0.55), assessing the motivation to play and enjoyment, increasing the internal consistency to α = 0.67 (N_complete_ = 95).

### 2.4. Statistics

To test whether both children and adults learn, we conduct a non-parametric one-sided Wilcoxon signed-rank test on the fraction learned for each group, testing whether it differed from zero. To test whether motor variability is higher following failure feedback than following success feedback, we conduct a non-parametric one-sided paired-samples Wilcoxon rank sum *t*-test on the median trial-to-trial ratio following non-rewarded trials and the median trial-to-trial ratio following rewarded trials for each group.

### 2.5. Deviations from Pre-Registration

We deviated from the pre-registration in two ways. The first deviation is that we used the self-reported motivation as our motivation measure. The second deviation is that we performed additional statistical tests to assess whether motor behavior developed from childhood to adulthood and whether motivation differed between groups, since we collected many participants in both groups. We compared the fraction learned and median trial-to-trial ratios between children and adults. We did so using one-sided Mann–Whitney U tests for testing whether children learned less than adults and whether they were more variable than adults. In addition, we tested whether the fraction learned increased with age from age seven to adulthood (ages 7–17 and all adults grouped at age 18) using a linear regression. We used a two-sided Pearson chi-square test for comparing motivations between children and adults.

## 3. Results

By visually checking the drawn trajectories, we concluded that participants were able to draw circle-like shapes without visual feedback of their hand or the trajectories they were drawing (see, for an example, Figure 1b). The median mean aspect ratio was 1.5 for children (IQR 1.3–1.9) and 1.3 for adults (IQR 1.2–1.5), respectively, and drawing path lengths were 9.8× the distance between start and endpoint of the drawings for children (IQR 7.8–12.7) and 12.1× that distance for adults (IQR 8.5–16.6). Participants took about 5.6 min (median, IQR 4.5–6.6) for completing the drawing task. The median success frequency was 0.47 (IQR 0.44–0.52), close to the intended success frequency of 0.5.

Children drew larger circles than adults, both during baseline and averaged across trials. At baseline, children drew with a median baseline radius of 2.8 cm (IQR 1.8–4.3) and adults 2.3 cm (IQR 1.3–3.2). Their overall median radii were 4.0 cm (IQR 2.4–4.9) for children and 2.9 cm (IQR 1.7–4.6) for adults.

### 3.1. Learning

As predicted, both children and adults adapted the radius of the circles based on the reward feedback. They increased the drawn radius over trials in the direction of the target radius (Figure 2a). Children showed a fraction learned of 0.43 (IQR 0.01–0.82, z = 5.4, *p* < 0.01) (Figure 2b). Adults showed a fraction learned of 0.66 (IQR 0.09–0.98, z = 5.4, *p* < 0.01) (Figure 2b).

### 3.2. Variability

Also in line with our prediction, reward feedback modulated the changes from trial to trial: following non-rewarded trials, participants changed their circle radius more than following rewarded trials (Figure 2c). Children showed higher trial-to-trial ratios following non-rewarded trials (median = 1.19, IQR 1.15–1.23) than following rewarded trials (median = 1.14, IQR 1.12–1.18) (z = 5.3, *p* < 0.01). Adults also showed higher trial-to-trial ratios following non-rewarded trials (median = 1.12 (IQR 1.09–1.16) than following rewarded trials (median = 1.08, IQR 1.07–1.10) (z = 4.5, *p* < 0.01).

### 3.3. Motivation

Self-reported motivation was the mean of the two items of the Intrinsic Motivation Inventory assessing enjoyment and motivation on a scale from low (1) to high (5) motivation. We observed a median mean motivation score of 4 (IQR 4–4.5) in children and 3.5 (IQR 2.5–4) in adults (Figure A2).

### 3.4. Additional Analyses

In addition to the pre-registered analyses, we explored how learning, variability, and motivation differed between children and adults, as well as how learning and variability evolved as a function of age. The fraction learned did not differ between children and adults (z = −1.0, *p* = 0.3; Figure 2b) and the linear regression analysis did not indicate any trend in learning from age seven to eighteen (b = 0.0, R^2^ = 0.01, *p* = 0.33; Figure 2d). Children were more variable than adults, both following success (y = 5.8, *p* < 0.01) and following failure (z = 4.7, *p* < 0.01; Figure 2c). The linear regression analysis indicated a downward trend in variability following reward from ages seven to eighteen (b = −0.007, R^2^ = 0.30, *p* < 0.01) and following reward absence (b = −0.008, R^2^ = 0.22, *p* < 0.01; Figure 2e). Lastly, adults were less motivated than children (*χ*^2^(8) = 25.2, *p* < 0.05; Figure A2).

By visual inspection, we observed no differences in the fraction learned between participants instructed in Dutch (48, of whom 28 children) and English (52, of whom 41 children) or between left- and right-handers, not when considering all participants and not in children or adults separately (Figure A3). We additionally explored task-irrelevant variability in parameters that do not influence the reward during the task: circle location and aspect ratio. We visually observed that children were more variable than adults also in the task-irrelevant variables (Figure A4). We also visually observed that both children and adults modulated task-irrelevant variability based on their successes and failures; the higher trial-to-trial changes in the radius, they also showed higher trial-to-trial changes in aspect ratio and circle location (Figure A4). In both age groups, task-irrelevant variability was similar for participants who were instructed in Dutch or English.

## 4. Discussion

The aim of this study was to test if children, like adults, exhibit reward-based motor learning. We developed a reward-based circle-drawing task with reward feedback based on the size of a drawn circle. Participants were able to perform the task: 98% met our criteria for drawing circles, and we managed to design a task that was motivating for children. It was, with children reporting higher motivation than adults. As predicted, both children (7–17 years old) and adults (21–53 years old) demonstrated motor learning by adapting the size of the circle they drew to the reward feedback they received. Furthermore, participants made larger changes in the size of the circle following reward absence compared to following reward presence, which is a hallmark feature of motor exploration [15]. Children did not learn less than adults and learning did not increase from age 7 to 17 (Figure 2e). Variability in the size of the drawn circle decreased from age 7 to 17 (Figure 2d).

We observed that children, like adults, demonstrate reward-based motor learning with an adaptive reward criterion that rewarded errors smaller than the median of the past ten trials. In the current study, they did so in a circle-drawing task that was carefully designed for children. The task was short, the one rewarded (task-relevant) variable was communicated explicitly, there was no feedback perturbation, and the narrative and graphics were designed to promote motivation in children. This finding adds to a recent report about an online study in which children aged 8–17 learned to adapt their reach direction based on binary reward feedback that was either deterministic or probabilistic. Our finding is furthermore consistent with another recent study [10] showing that children aged 6–12 can learn a throwing task based on categorical reinforcement feedback (success, too far, or too short) based on a fixed reward criterion.

Although adults could be expected to learn more due to their more developed nervous system and musculoskeletal system, we did not find that children (7–17) learn less than adults (18–65). Consistently, a regression analysis showed no increase in learning between ages 7 and 17 (Figure 2d). This result contrasts with the recent results of [9], which found that reward-based learning was already present at age 3 and increased with age until it reached adult-like learning around the age of 17. Possibly, the different results are due to the younger study population in Hill et al. (3–6 years old), who showed the most pronounced difference with adults. However, in our study, already for the 7 year olds, learning was comparable to adult learning (Figure 2d). The different conclusions might be due the variability in the measurement of learning. Although both studies used a large sample size (69 children in our study, and 81 children in the most comparable experimental condition of [9]), reward-based learning is inherently variable between individuals, as it is a stochastic process that depends on random exploration. Alternatively, although both experimental tasks have been designed for children, our task might have been slightly more intuitive for children. Hill and colleagues [9] used a reaching task, based on the visuo-motor rotation paradigm [34] in which children helped a penguin cross an icy river with forward movements on the trackpad or with the mouse. Reward was based on the position where the penguin crossed the river. To start the next trial, a return movement had to be made for which no feedback was provided. Our task might have been slightly more intuitive, as the circle naturally returned the hand to the starting location for the next circle and could be performed without experimenter or parental help. Also, we used auditory feedback in addition to the visual feedback, and the digital pen might have been easier to control than the computer mouse or trackpad.

Both children and adults used the reward feedback to explore which circle size led to a reward, indicated by their increase in variability following failure as compared to following success. Although exploration could be formalized as the difference in variability following non-rewarded and rewarded movements, in the current study, we did not quantify exploration due to the low number of trials [30]. Hence, while we can compare the variability of the children to that of the adults, we cannot compare exploration. Nonetheless, children (and adults) seem to explore and (resultantly) learn. Interestingly, both variability following success and following failure decreased with age, which is in contrast to the results of Hill and colleagues [9], who found an increase in variability following failure and a decrease in variability following success. This might be related to differences between tasks in terms of the number of targets, the information available during movement, or to differences in the measurement of variability. Since variability is generally interpreted in relation to learning, with learning being related to the ratio between exploration and motor noise [13,14], the meaning of the different trends for learning is hard to interpret.

Humans explore both task-relevant and task-irrelevant dimensions during reward-based motor learning [35]. Other work suggests that children’s exploration might be less well-aligned to the relevant task dimension than adults’ exploration (for a review, see [27]). Using a task in which participants controlled the lateral position of a visual cursor with the movements of their upper body and learned the mapping between upper body angle and lateral cursor position, it has been found that children were slower to align their variability to the task-relevant dimension [28]. Our study provided no indication that children and adults differentially align their variability to the task-relevant dimension, as they appeared to increase variability in the task-irrelevant dimensions of circularity and circle position similar to how adults increased the variability in these dimensions (Figure A3).

To assess variability more reliably and to compare exploration between age groups, future studies will have to include more trials. We previously recommended measuring about 500 trials [30]. The 80–100 trials were measured in, on average, 6 min. After these 6 min, children still reported high motivation. Hence, within twenty minutes, the trial number could be increased sufficiently.

We conclude that our circle task is suited for assessing reward-based motor learning in children (7–17). Within a 6-min task, we found substantial reward-based motor learning across ages (Figure 2a,b) and observed the hallmark of exploration: a clear increase in variability following failure for both age groups (Figure 2c). Some advantages of our task are that the next movement starts close to where the previous movement ends and that the complexity of the task can be easily increased by rewarding additional shape aspects such as smoothness and symmetry rather than size only. This will allow future studies to test whether differences between children and adults arise when the task is more complex. For instance, children might have more problems in solving the credit assignment problem when the feedback is based on multiple aspects of the drawn trajectory. The set-up using a laptop and Wacom device is portable and does not require participants to visit the lab, which might be an obstacle when testing children [12].

Our study limitations can be improved by adding trials with performance-independent feedback (‘error clamps’ [36]) to study the modulation of variability following success and failure [9]. Also, withholding feedback or providing a reward clamp in the first five trials will help to establish a more reliable baseline. Finally, the measurement of motivation can be improved: the internal reliability of the questionnaire items was low, and experimenters had trouble facilitating the proper behavioral measurement of motivation. In future studies, the behavioral measure of motivation might be implemented in the task by increasing the number of trials in the motivation phase and removing the verbal experimenter instructions before the motivation phase. However, even without these adjustments, the task we established can be used for measuring reward-based motor learning in children.

## Figures and Tables

**Figure 1 behavsci-14-01055-f001:**
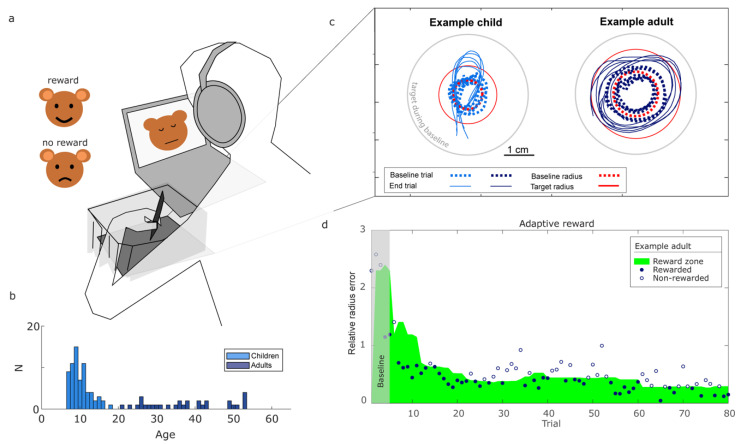
Experimental task. (**a**) Participants were presented with a picture of a noseless bear on a laptop monitor. Their instruction was to give the bear a nose by drawing a circle of a correct size on a drawing tablet next to the laptop. Circles were drawn without visual feedback of the hand and drawing trajectory. After each trial, binary reward feedback was provided by means of a happy or sad bear face and a bell or buzzer sound. (**b**) Histogram of the ages in our study population. (**c**) Examples of circles drawn by one of the children and one of the adults. The first and the last five attempts are shown, as well as a circle with the individual baseline radius (dashed circle) and one with the target radius (solid circle), which was set to twice the baseline radius. (**d**) Trials were adaptively rewarded if errors were smaller than the fifth-sized error of the past ten trials. This was done to keep the reward frequency close to 50%.

**Figure 2 behavsci-14-01055-f002:**
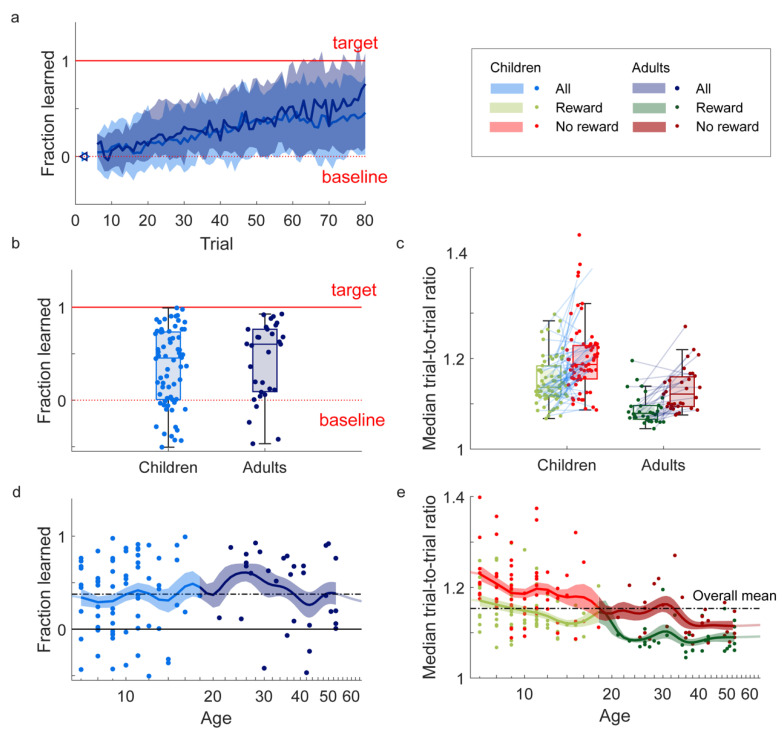
Learning and variability during the task. (**a**) Fraction learned over trials. Plotted are the median and interquartile range over the participants. (**b**) Final fraction learned per age group. (**c**) Variability following rewarded (signaling success) and non-rewarded (signaling failure) trials, expressed as the median trial-to-trial ratio in the radii. (**d**) Final fraction learned as a function of age. Note that age has been plotted on a logarithmic scale. For visualization purposes, we computed smoothed curves of the development of learning as a function of log-transformed age by calculating weighted averages using a moving Gaussian with a standard deviation of log(1.1) for defining weights of data points centered around each log-transformed year of age [33]). (**e**) Variability following success (in grayscale print: lighter gray) and following failure (in grayscale print: darker gray) as a function of age. Same method as panel (**d**). In all panels, dots indicate individual participants.

## Data Availability

Data and protocol can be found at https://osf.io/qtn3k/ (folder Data January 2023, accessed on 8 October 2024).

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
