# Peer review of "A Circle-Drawing Task for Studying Reward-Based Motor Learning in Children and Adults"

_behavsci, 2024, doi:10.3390/bs14111055_

Round 1

Reviewer 1 Report

Comments and Suggestions for Authors

This manuscript is well-written, scientifically valid, technically sound and can make an original contribution to the literature as it is concerned with the designing of a new ‘circle-drawing’ task suitable for assessing reward-based motor learning in both children and adults, an issue that is of great value  in motor learning literature.

Although the manuscript seems to be well-structured there are some minor points in the introduction section that need attention.

·        e.g. In line 38, the aim of the study”, line 59, “we expect…”, line 69, “While we expect children to show…” . The aim/aims of the study and research questions are mentioned in different paragraphs, and this may be confusing for the reader. It is recommended that the authors should keep their introduction brief  and should organize it according to research hypotheses and aims which are mentioned in the text. However, the aim/aims of the study should be summarized in the last paragraph of the introduction section in order to become more clearly stated for the reader.

·        In lines 81-87 there is a description of the task. It is advisable that this description should be more brief in the introduction section and should be better included in the materials and methods section.

·        The use of past tenses is recommended when studies conducted in the past are described (e.g. line  81, “The aim of this study is …” it is better to write “The aim of this study was …” and elsewhere in the text). Moreover, it is advisable the use of 3rd person and passive voice, all over the manuscript, when is needed (e.g. lines 82-83, “To make sure that we could measure reward-based motor learning in children, 82 we developed a reward-based circle-drawing task with reward feedback based on the size…”, instead “To make sure that reward-based motor learning we could be measured  in children, a reward-based circle-drawing task was developed with reward feedback based on the size…”). In this way the text becomes more neutral.

Author Response

This manuscript is well-written, scientifically valid, technically sound and can make an original contribution to the literature as it is concerned with the designing of a new ‘circle-drawing’ task suitable for assessing reward-based motor learning in both children and adults, an issue that is of great value  in motor learning literature.

We thank the reviewer for their positive assessment of the scientific quality and relevance of our study.

Although the manuscript seems to be well-structured there are some minor points in the introduction section that need attention.

  • e.g. In line 38, the aim of the study”, line 59, “we expect…”, line 69, “While we expect children to show…” . The aim/aims of the study and research questions are mentioned in different paragraphs, and this may be confusing for the reader. It is recommended that the authors should keep their introduction brief  and should organize it according to research hypotheses and aims which are mentioned in the text. However, the aim/aims of the study should be summarized in the last paragraph of the introduction section in order to become more clearly stated for the reader.

We agree that this is confusing. We’ve made a textual adjustment in line 42, rephrasing “the aim of this study is to” to “it is important to”. This way, the aim is now only in the last paragraph.

  • In lines 81-87 there is a description of the task. It is advisable that this description should be more brief in the introduction section and should be better included in the materials and methods section.

We agree with you and moved the task description and its benefits to section 2.2 Procedure and experimental task.

  • The use of past tenses is recommended when studies conducted in the past are described (e.g. line  81, “The aim of this study is …” it is better to write “The aim of this study was …” and elsewhere in the text). Moreover, it is advisable the use of 3rd person and passive voice, all over the manuscript, when is needed (e.g. lines 82-83, “To make sure that we could measure reward-based motor learning in children, 82 we developed a reward-based circle-drawing task with reward feedback based on the size…”, instead “To make sure that reward-based motor learning we could be measured  in children, a reward-based circle-drawing task was developed with reward feedback based on the size…”). In this way the text becomes more neutral.

Thank you for suggesting improvements to our writing style. We have changed the past tense based on your suggestions but prefer the active voice. In our field, this is a common way of writing which is for instance advocated in Mensh & Kording (2017).

Mensh B, Kording K (2017) Ten simple rules for structuring papers. PLoS Comput Biol 13(9): e1005619. https://doi.org/10.1371/journal.pcbi.1005619

Reviewer 2 Report

Comments and Suggestions for Authors

Please add more information about participants to the abstract, such as age mean, gender, ….

You called the age range of 7 to 17 years children. Is it really so? A child is usually up to 11 years old, and a 17-year-old person is an adolescence and not a child. Please consider this in the title and the entire manuscript.  Also, for adults (as you said, 18-65 years), a person around 60 years old is old adult but not adult.

Statistical indices and significance levels should be mentioned in the abstract for all comparisons.

I think the introduction is well written. It is only suggested to mention some hypotheses at the end of the introduction.

On what basis was the sample size calculated? Have you used software such as GPower?

Please add the inclusion and exclusion criteria.

Were the instructions and how they were presented the same for both age groups?

The discussion was well written, but please add the limitations of the research at the end of the discussion.

Author Response

We thank the reviewer for the time invested in commenting on our manuscript.

Please add more information about participants to the abstract, such as age mean, gender, …

We have added age mean and standard deviations to the abstract. For gender, we chose not to do so, since we do not regard gender as relevant for the research question and also do not separately analyse gender groups in the paper.  

You called the age range of 7 to 17 years children. Is it really so? A child is usually up to 11 years old, and a 17-year-old person is an adolescence and not a child. Please consider this in the title and the entire manuscript.  Also, for adults (as you said, 18-65 years), a person around 60 years old is old adult but not adult.

As reward-based motor learning has been virtually exclusively studied in an age group from 18 years and up, our aim was to test reward-based motor learning in ages younger than 18 years old. We hence used the term ‘children’ to refer to ‘non-adults’. Indeed, from developmental perspective childhood is up to 11 years old and adolescence is up to 17 years old. We have clarified our definition in the Introduction’s first paragraph and have added more details on the mean age of the age groups in the abstract.

Statistical indices and significance levels should be mentioned in the abstract for all comparisons.

We wrote the paper according to the submission guidelines and with a writing style that is common in our field. As described in Mensh & Kording (2017), we focused on transferring the main conclusions of the paper in the abstract, without many details. As the p-value has been designed for a true or false judgement on the hypothesis and not as a continuous measure of evidence, we prefer not to mention p-values in the abstract. Low p-values may falsely cause the reader to believe we gathered particularly strong evidence. We therefore prefer to keep the abstract as it was, without statistical indices and significance levels.

Mensh B, Kording K (2017) Ten simple rules for structuring papers. PLoS Comput Biol 13(9): e1005619. https://doi.org/10.1371/journal.pcbi.1005619

I think the introduction is well written. It is only suggested to mention some hypotheses at the end of the introduction.

Thanks! In the revised version of the manuscript, we explicitly mention our hypothesis at the end of the Introduction. The specific predictions we made are at the end of the introduction.

On what basis was the sample size calculated? Have you used software such as GPower?

Although not with GPower, we estimated the sample size we would need for measuring reward-based motor learning in children to be 50 children. We based this on student pilot results and our earlier reward-based motor learning studies. From our previous studies we know how variable this type of learning is. To improve statistical power, we therefore decided to maximize the number of participants by designing a short experiment and measuring two participants at the same time, and to minimize the number of tests performed on the data. Referring to your first comment, this is also why we did not separately analyze different age groups within the group of children.

Please add the inclusion and exclusion criteria.

Participants were included based on age and communication language and the presence of a parent or legal guard to sign their consent form (Methods, line 117). We made the inclusion criteria more explicit by stating that apart from age 7 to 65, participants younger than 18 had to bring their parent or legal guard for signing the informed consent forms.

Were the instructions and how they were presented the same for both age groups?

You were not the only one to raise this question. We now clarified at the start of the procedure section (2.2, line 151) that experimenters followed a script with written instructions to make sure that all participants (also children and adults) were instructed in the same way. We added the protocol to the Open Science Foundation weblink.

The discussion was well written, but please add the limitations of the research at the end of the discussion.

In fact, the current last paragraph already contains the limitations of the study, but we’ve made a textual adjustment on line 444 (“Our study limitations can be improved “) to make this more explicit.

Reviewer 3 Report

Comments and Suggestions for Authors

Dear authors,

Thank you for the opportunity to review your interesting and well-executed study on reward-based motor learning in children. I appreciate the time and effort your team spent thoughtfully designing the experiment and addressing a gap in developmental motor learning. That said, there are a few areas where further clarification and expansion could strengthen the manuscript. Below, I have outlined specific questions to help guide potential revisions.

Introduction:

  1. When discussing motor variability in children, you link it to motor noise. Could you please provide more detail on how this variability specifically influences learning? Is motor variability always a challenge for learning, or could it sometimes play a beneficial role in facilitating exploratory learning behaviors?
  2. You state that children’s brains and musculoskeletal systems still develop, affecting motor learning. Could you expand on how these developmental differences, particularly in neuroplasticity and motor control, impact the motor learning process in children compared to adults?
  3. The introduction treats children as a fairly homogenous group, but there is likely significant variability within the child population. Have you considered how factors such as age, prior motor experience, or even cognitive development might influence motor learning outcomes? How did you account for these individual differences in your study?
  4. You mention that task complexity negatively affects learning, particularly in children. Could you elaborate further on why task complexity impedes reward-based motor learning, and whether there is any evidence that certain types of complexity might be more challenging than others?
  5. The circle-drawing task is interesting, but what was your rationale for choosing this task as the primary means of assessing motor learning in children? Could you explain why this task is especially appropriate for children compared to other potential motor learning tasks?
  6. You reference several studies involving adults when discussing reward-based motor learning. Given that your study focuses on children, would it not strengthen your argument to incorporate more child-specific research? Could you discuss any relevant studies on motor learning in children to better support your claims?

Materials and Methods:

  1. Thank you for your acknowledgement of deviations from your pre-registered protocol, particularly the switch from a behavioral measure of motivation to a self-reported one. Given that the self-reported measure had a low Cronbach’s alpha (α = 0.56), could you explain in greater detail why this measure was still chosen? What was the rationale for ultimately analyzing only two questions from the Intrinsic Motivation Inventory, and how do you make sure that this does not lead to concerns about data manipulation or cherry-picking?
  2. You excluded 28 datasets due to technical issues or incomplete tasks. Were these exclusions systematically checked for any patterns, such as differences in age, handedness, or language group? Could you justify the exclusion of these datasets and discuss whether these exclusions might have affected the internal validity of the study?
  3. The task design, which involves drawing circles without visual feedback, seems rather simplistic. Why was this task chosen as the primary motor learning task for both children and adults? How do the outcomes of this task relate to more complex, real-world motor learning scenarios, and what was the rationale for focusing on circle size as the primary variable?
  4. The motivation phase seems to have suffered from inconsistent instruction across participants, and the results show a bimodal distribution of responses, where participants either completed 0 trials or 20 trials. How do you interpret these results, and what does this say about participant understanding or engagement with the task? Would standardizing instructions have improved the consistency of this phase?

Author Response

Dear authors,

Thank you for the opportunity to review your interesting and well-executed study on reward-based motor learning in children. I appreciate the time and effort your team spent thoughtfully designing the experiment and addressing a gap in developmental motor learning. That said, there are a few areas where further clarification and expansion could strengthen the manuscript. Below, I have outlined specific questions to help guide potential revisions.

We thank the reviewer for appreciating our study, and for the detailed questions and comments provided.

Introduction:

  1. When discussing motor variability in children, you link it to motor noise. Could you please provide more detail on how this variability specifically influences learning? Is motor variability always a challenge for learning, or could it sometimes play a beneficial role in facilitating exploratory learning behaviors?

We agree that it is important to be specific about motor variability. In the introduction, we spent lines 77-84 explaining that literature shows that low motor noise and high exploration facilitate learning.

“In addition, motor control is important in reward-based motor learning: highly precise movements due to low motor noise variability, and high variability due to exploration both facilitate reward-based motor learning [13, 23]. Children have demonstrated higher levels of motor variability, which might be attributed to motor noise [24-26], and might hence show reduced reward-based learning. Moreover, children have been shown to tune variability less to the task-relevant dimensions where exploration is beneficial [27, 28], which might hamper their reward-based learning as compared to adults.”

  1. You state that children’s brains and musculoskeletal systems still develop, affecting motor learning. Could you expand on how these developmental differences, particularly in neuroplasticity and motor control, impact the motor learning process in children compared to adults?

We address this question in the third paragraph of the introduction (line 73), where we mention how the timing of maturation of the basal ganglia and prefrontal cortex may influence motor learning. We have now added a sentence (line 82) on the possible role of the fast-changing musculoskeletal system of children as compared to adults. As we are not aware of studies showing a relationship between neuroplasticity and reward-based motor learning, we have chosen not to discuss neuroplasticity.

  1. The introduction treats children as a fairly homogenous group, but there is likely significant variability within the child population. Have you considered how factors such as age, prior motor experience, or even cognitive development might influence motor learning outcomes? How did you account for these individual differences in your study?

We agree that children are a non-homogeneous group in terms of age, motor experience and cognitive development. Importantly, when we were designing our study, no data on reward-based motor learning were available. As a first step, we aimed to develop a motor learning task that could be used by any child older than 7, and test whether reward-based motor learning occurs in children as a homogeneous group, too. Based on our experience with measuring the highly variable process of reward-based motor learning, we decided to maximize statistical power, by maximizing the number of participants per group and to minimize the number of statistical tests to perform.

Whereas we did not analyze how factors such as age, prior motor experience and cognitive development may influence motor learning outcomes, we did consider these factors in the design of the task. We designed a task that does not heavily rely on prior motor experience or cognitive development. It was meant to be understandable and doable for children aged 7 and higher. As we agree with you that children are not a homogeneous group, we did explore how learning is related to age (see Figure 2d, e).

  1. You mention that task complexity negatively affects learning, particularly in children. Could you elaborate further on why task complexity impedes reward-based motor learning, and whether there is any evidence that certain types of complexity might be more challenging than others?

In the introduction (line 61-63), we have clarified why task complexity might impede reward-based motor learning: “While simple tasks can be learned in this way, the effectiveness of reward-based motor learning decreases with task complexity as the mapping between control parameters and feedback becomes many-to-one, requiring the learner to solve a credit-assignment problem.”

In the discussion paragraph we further propose that task complexity might especially impede learning in children as they might have greater difficulty in solving the credit assignment problem.

  1. The circle-drawing task is interesting, but what was your rationale for choosing this task as the primary means of assessing motor learning in children? Could you explain why this task is especially appropriate for children compared to other potential motor learning tasks?

We would like to refer to the answer to comment 3 below.

  1. You reference several studies involving adults when discussing reward-based motor learning. Given that your study focuses on children, would it not strengthen your argument to incorporate more child-specific research? Could you discuss any relevant studies on motor learning in children to better support your claims?

When we designed our study, no data of reward-based motor learning in children were available. Therefore, we took a first step by testing whether children show reward-based learning. Simply for this reason, we cannot add extra relevant studies on reward-based motor learning in children.

Materials and Methods:

  1. Thank you for your acknowledgement of deviations from your pre-registered protocol, particularly the switch from a behavioral measure of motivation to a self-reported one. Given that the self-reported measure had a low Cronbach’s alpha (α = 0.56), could you explain in greater detail why this measure was still chosen? What was the rationale for ultimately analyzing only two questions from the Intrinsic Motivation Inventory, and how do you make sure that this does not lead to concerns about data manipulation or cherry-picking?

You are right that the Cronbach’s alpha is low. We therefore reported that we found this value and that we found it low (2.3.3 Motivation, line 268-290). We explained (2.3.3 Motivation, line 268-290) that we therefore opted to use the best-available measure of motivation by picking the highest correlated two motivation items.

Since the analysis on motivation was not used for answering our main question, we believe that we could explore motivation this way. We also listed the motivation measurement as a limitation of our study in the last paragraph of the discussion (Discussion, lines 447-452).

  1. You excluded 28 datasets due to technical issues or incomplete tasks. Were these exclusions systematically checked for any patterns, such as differences in age, handedness, or language group? Could you justify the exclusion of these datasets and discuss whether these exclusions might have affected the internal validity of the study?

We have clarified the justification of the exclusion in the revised methods paragraph by making explicit that the technical problems referred to a software bug which caused computer crashes (line 161). The exclusions could not have affected the internal validity, because only two exclusions (on a total sample of 128) were based on the behavior of the participant. The other 26 exclusions were based on a software problem that was not related to the participant’s behavior.

  1. The task design, which involves drawing circles without visual feedback, seems rather simplistic. Why was this task chosen as the primary motor learning task for both children and adults? How do the outcomes of this task relate to more complex, real-world motor learning scenarios, and what was the rationale for focusing on circle size as the primary variable?

Our study is the first to study reward-based motor learning in children. To not make too big steps, we designed a task that was very similar to the paradigms used for studying this type of learning in adults. These paradigms are very specific and rather simplistic: participants generally make repeated center-out target-directed movements without vision of their moving hand. Whereas at the start of an experiment, participants get rewarded when their hand crosses an invisible reward zone that is centered around the visual target, over trials, this invisible reward zone is moved away from the visual target. This way, participants need to adjust their movements to this feedback perturbation, allowing researchers to study learning. We chose to make the task suitable for children by adding a storyline, removing any feedback perturbations (i.e. making it a bit more real-world-like) and allowing participants to start their movement at a location of preference. We focused on circle size as the primary variable as we considered size a concept that could easily be understood by participants aged 7 and older.

  1. The motivation phase seems to have suffered from inconsistent instruction across participants, and the results show a bimodal distribution of responses, where participants either completed 0 trials or 20 trials. How do you interpret these results, and what does this say about participant understanding or engagement with the task? Would standardizing instructions have improved the consistency of this phase?
  • Instructions were standardized as experimenters followed a script with verbal instructions. We now clarified this at the start of the procedure section (2.2, line 151).
  • Based on feedback of the experimenters that they found it hard to believably pretend that they were busy while participants could perform the motivation phase of the experiment, we do however agree that instruction may have been inconsistent. We believe that this was not due to a lack of standard instructions, but due to insufficient training of how to transfer the verbal instructions, focusing on intonation and acting to be busy filling in forms during the break.
  • We interpret the bimodal distribution as the motivation phase measuring a binary decision to continue or not. Our measure was thus not sensitive enough to test for small differences in motivation between participants. It had a low correlation with self-reported motivation measures. We are uncertain what this says about participants’ engagement with the task. We do not believe that the bimodal distribution arose because participants did not understand the instruction before the motivation phase. Whether this belief is correct, may be tested in a new experiment with a longer motivation phase. This we also mention in the limitations paragraph of the discussion Discussion (lines 447-452).

Round 2

Reviewer 3 Report

Comments and Suggestions for Authors

Dear authors,

Thank you for your detailed revisions and thoughtful responses to the reviewer’s comments. Your study contributes to understanding reward-based motor learning in children, and the revisions have significantly strengthened the manuscript.

There are just a few minor points for clarification that could be addressed as final adjustments:

Handedness:

It would be helpful to briefly mention handedness in the methods or results section. Even if handedness did not have a significant impact, a short statement acknowledging that it was considered or did not influence the outcomes would provide additional reassurance against potential bias.

Exclusion Justification:

You’ve done a good job clarifying the exclusions due to technical issues. A quick confirmation that these exclusions were not biased by patterns such as age or handedness would further solidify the integrity of your sample.

Task Complexity:

The explanation of task complexity is well-covered. Including a brief note on whether cognitive or motor complexity presents distinct challenges for children would add a additional layer to your discussion.

Author Response

Dear authors,

Thank you for your detailed revisions and thoughtful responses to the reviewer’s comments. Your study contributes to understanding reward-based motor learning in children, and the revisions have significantly strengthened the manuscript.

There are just a few minor points for clarification that could be addressed as final adjustments:

Dear reviewer,

We have added the details to the manuscript according to your wishes.

Thanks.

Handedness:

It would be helpful to briefly mention handedness in the methods or results section. Even if handedness did not have a significant impact, a short statement acknowledging that it was considered or did not influence the outcomes would provide additional reassurance against potential bias.

We report handedness in the methods section: “Of the 128 participants who participated, we included 100 participants in the data analysis: 69 children (28 male, 37 female, 4 other; 10 left-handed; reported rounded-down age [29] 10.1±2.5 (mean±SD) years) and 31 adults (18 male, 12 female, 1 other; 5 left-handed; reported rounded-down age 37.6 ± 10.2 years) (Figure 1b).” As the group of left-handers is too small to compare to the much bigger group of right-handers, we did not perform statistical tests but we visually we could observe no pattern of a difference between left- and right-handers. We have added a sentence on line 352 to address this.

Exclusion Justification:

You’ve done a good job clarifying the exclusions due to technical issues. A quick confirmation that these exclusions were not biased by patterns such as age or handedness would further solidify the integrity of your sample.

Thanks. We cannot confirm this, since the data that we would need for such a confirmation haven’t been recorded. Age and handedness data were collected in the questionnaire, which participants completed following the drawing task. If the drawing task crashed, it did not make sense for participants to complete the questionnaire. Hence, we cannot report any patterns. What we have added to the manuscript though, is that the computer crashes were independent of participants drawing behaviour and that the datasets were not complete and therefore not included in the analysis. We have no reason to assume that in the specific hours of the computer crashes, a different sample of people was in the museum than in the rest of the days, and therefore strongly believe that the exclusion does not threaten the integrity of the sample. Moreover, the age and handedness data of the sample that we have collected are broad in terms of age and in line with the +-10% left-handers in the population.

Task Complexity:

The explanation of task complexity is well-covered. Including a brief note on whether cognitive or motor complexity presents distinct challenges for children would add a additional layer to your discussion.

We like this question! As complexity is not the focus of the current study, we think discussing this further in the manuscript is beyond the scope of the manuscript.
